# Keeping Sane in a Changing Climate: Assessing Psychologists’ Preparedness, Exposure to Climate-Health Impacts, Willingness to Act on Climate Change, and Barriers to Effective Action

**DOI:** 10.3390/ijerph21020218

**Published:** 2024-02-13

**Authors:** Gabriela Stilita, Fiona Charlson

**Affiliations:** 1School of Public Health, Faculty of Medicine, University of Queensland, Herston, QLD 4006, Australia; f.charlson@uq.edu.au; 2Queensland Centre for Mental Health Research, Queensland Health, Wacol, QLD 4076, Australia; 3Psychology Department, The Prince Charles Hospital–Queensland Health, Chermside, QLD 4032, Australia

**Keywords:** climate change, mental health, psychologists, professional’s role, health impacts, professional preparedness

## Abstract

Evidence of the impact of climate change on mental health is growing rapidly, and healthcare professionals are being called to be active participants in protecting the population’s health. Yet, little is known about psychologists’ understanding of climate-health impacts and their role in mitigation actions. We surveyed Australian psychologists (*N* = 59) to examine preparedness in identifying and managing the impact of climate change on mental health, exposure to climate-health impacts, willingness to act, and barriers to acting on climate change. Data was analysed through descriptive and associative methods. We found that participants are not prepared to identify and manage mental health presentations related to climate change, and they are not engaged in climate change mitigation. We identified that a lack of knowledge of climate-health impacts and tackling and mitigation strategies, in addition to ethical concerns, were the main barriers to engagement with communication and advocacy. With the impacts of climate change on mental health expected to soar, there is a clear and urgent need to prepare the psychological workforce to address this public health issue by establishing professional education programs and reframing climate change as a health crisis.

## 1. Introduction

Climate change impacts mental health through a complex multidimensional interaction between causal and correlational pathways [1,2]. Directly or indirectly, climate change is linked to post-traumatic stress disorder (PTSD), anxiety, depressive symptoms, aggressive behaviour, domestic violence, behaviour changes in children, psychiatric hospitalisation, suicidality, substance use disorders and medication adverse side effects due to changes in metabolic processing [1,3]. Awareness of natural degradation and the predicted impacts of climate change decreases mental well-being and increases the incidence of depressive and anxiety symptoms and pre-traumatic stress [4,5]. Furthermore, most of the well-established climate-related impacts on general health are linked with a higher incidence of mental health issues, as mental health conditions are common among those experiencing cardiovascular disease [6], kidney diseases [7], allergies, asthma, and lung diseases [8,9]. 

When we consider the complex interaction between socioeconomic determinants of health and exposure to climate change, it becomes clear that the full impact of climate change on people’s mental health has not yet been fully understood. It has been argued that only the adoption of a systems thinking approach [10] will make it possible to identify the intricate relationship between various vulnerability factors and different exposure pathways that result in poorer mental health caused by climate change [2,11,12]. As climate change worsens, it is sensible to expect increases in mental health presentations as a result of the increased incidence of physical health conditions caused by climate change, including as a result of increased awareness of environmental degradation, forced displacement, community disruption, nutritional depletion, and increased poverty, among others. Thus, with the escalation of climate change, demand for mental health services is expected to grow [3].

Notwithstanding the severe impacts on health caused by climate change, the perceived role of health professionals in climate change has not been the subject of many studies. Evidence shows that although health professionals agree that climate change impacts health, they have insufficient information on the link between climate change and health [13,14]. 

The healthcare sector is a huge contributor to climate change. Directly or indirectly, the sector is responsible for between 1% and 5% of the world’s greenhouse emissions [15]. The healthcare sector footprint has triggered recent calls for a paradigm shift in how we understand health; the inclusion of environmental health into healthcare education, and the healthcare professional’s responsibility to address how they relate to climate change in their professional practice [16,17,18]. The planetary health paradigm understands the direct link between human health and environmental health, framing climate change as a planetary as well as a human health crisis [18,19,20]. The shift towards the planetary health paradigm and the urgent need to address climate change and its forecasted pressure on healthcare systems has led the World Health Organisation to recently publish its framework for a climate-resistant and low-carbon health system, highlighting the need for a climate-smart health workforce [21]. 

Environmental disasters are common in Australia. Although it is hard to estimate the full impact of climate change on Australians’ mental health, recent epidemiological data on the prevalence of mental health issues related to climate change show that 25% and 16·6% of respondents met screening criteria for PTSD and pre-traumatic stress, respectively, while about 9% were significantly impacted by eco-anxiety [5]. Another study found that, among Australians who experienced extreme weather events, 73% reported anxiety symptoms, 49% reported depressive symptoms, 30% had symptoms of PTSD, and 25% had their existing mental health conditions exacerbated [22]. 

A recent national survey conducted by the Climate and Health Alliance (CAHA), which assessed 875 Australian healthcare professionals’ (specifically physicians, nurses, midwives, public health professionals and medical students) insights into climate change reported similar results from previous studies [14,15,23]. To the best of our knowledge, there is no research on psychologists’ perceptions of their professional role in climate change, although professional psychological associations have published resolutions and guidelines that acknowledge the unique contribution psychologists can have to climate change mitigation and adaptation [24,25,26]. However, their efficiency in fulfilling their purpose is unknown. Data on psychologists’ understanding of climate-health impacts and their professional role in mitigating climate change are non-existent. Thus, the present exploratory study aims to identify and analyse factors associated with Australian psychologists’ preparedness to identify and manage: (1)The impact of climate change on mental health,(2)Exposure to climate-health impacts,(3)Willingness and barriers to acting on climate change.

## 2. Materials and Methods

### 2.1. Study Design

We conducted an online survey using an adapted version of the survey RUN, performed by the Climate and Health Alliance (CAHA), which assessed Australian healthcare professionals’ insights into climate change [23]. The original CAHA survey did not include psychologists. In addition, psychologists often commit longer time to direct contact with their clients than the health professionals surveyed previously. The choice of a cross-sectional survey was justified by its capacity to access a large number of participants and compare the results of the two studies. After making the necessary adaptations, we piloted the questionnaire with five psychologists experienced in climate change and its impact on mental health, resulting in language modification and the removal of repetition.

The final questionnaire consisted of 39 questions (two open questions and 37 closed questions, of which nine had an “other” option for further information). The questionnaire was divided into three sections. The first section focused on demographic data to gather information about age, region of residence, registration type and years of experience, and also included workplace-related questions (related to decision-making capacity, client-facing roles and work settings). The second section consisted of questions related to personal views on climate change, personal experience with climate change and the impact of climate change in the workplace. The last set of questions examined the participants’ perception of their preparedness to identify and manage the health impacts of climate change and asked their opinion about the claim that their professional role includes tackling climate change. A copy of the survey can be found in the Appendix A. 

### 2.2. Participants

Our only inclusion criterion was current registration as a psychologist with the Australian Health Practitioner Registration Agency (AHPRA). There are three types of registration as a psychologist in Australia: provisional registration, granted to students in their final years of study, in which the student can practice under strict supervision in authorised placements; general registration, granted to those that have successfully completed six years of psychological studies; and general registration with endorsements, which is granted to those who have completed a specialised Master degree, followed by two years of intense supervision [27]. We decided to include the three types of registered psychologists, as they all have access to clients in their professional practices. Between July and September 2022, there were 35,315 general registered psychologists and 7977 provisional psychologists, totalling 43,292 professionals of interest [28]. 

Participants were recruited through social media posts, personal contacts, and communications conducted through professional associations. The recruitment ad was featured in the Australian Psychological Society (APS) weekly newsletter on 14 October 2022 [29], and in the CAHA weekly newsletter on 28 September 2022 [30]. The number of psychologists who had access to the recruitment ads is unknown. Every respondent who started the survey finished it, resulting in a 100% completion rate. 

Data were collected via Qualtrics, over a nine-week period between September and November 2022. 

### 2.3. Variables

To assess preparedness, we based our analysis on the competencies knowledge of the discipline, intervention strategies and research and evaluation (as established by the Psychology Board of Australia (PBA) [27]), focusing on professional experience, professional endorsement, personal interest in climate change, personal importance attributed to climate change, perceived preparedness, previous knowledge of climate change, and previous training related to climate change. For exposure to climate impacts on health, we looked at exposure to climate-driven events, personal emotional impact, clients presenting with climate-related health impacts and work being affected by climate change in the past twelve months. For willingness and barriers to acting on climate change, we were interested in personal interest in and personal importance attributed to climate change, ethics, decision-making role in professional practice, perceived professional responsibility in addressing climate change, and perceived enablers and barriers related to tackling climate change at a professional level.

It is important to mention that, unless otherwise specified, the questions on the questionnaire focus on the health impacts of climate change, whatever these impacts are. This decision is justified by our holistic understanding of health. In our view, the strict division of health into physical or mental health is counterproductive, and is not evidence-based. 

### 2.4. Statistical Analysis

Statistical analysis was performed using Excel for data cleaning and R commander version 2·7-2 for Mac—https://www.R-project.or/ R Foundation for Statistical Computing, Austria (accessed on 31 July 2022) was used for data analysis. Quantitative data were summarised as frequencies, and a descriptive analysis of all variables was calculated before associations between variables were carefully explored. Given the small sample size, some questions with Likert scales required data to be combined into two nominal categories to perform a chi-square test for the statistical analysis. Participants with missing data were kept in the sample, as they were presented during descriptive analysis, and did not influence our results. Three participants declared they did not directly work with clients: One did not answer the question, one works exclusively in academia, and the other is a researcher. After careful consideration, we opted to keep the three professionals in the statistical analysis when assessing preparedness, because their current registration as a psychologist allows them to have direct client contact whenever they wish. Moreover, one can only incorporate any discussion or teaching of the impact of climate change on mental health if one is prepared for it. 

Qualitative data gathered by participants’ responses to the “other” questions are presented to illustrate the points raised by participants, as there are not enough answers to justify a complete qualitative analysis. 

The study was approved by the University of Queensland Human Research Ethical Committee (2022/HE001135).

## 3. Results

### 3.1. Sample Characteristics

We had 61 respondents, but two were excluded as they were not registered psychologists. The final sample consisted of 59 participants, predominantly female, aged between 35–44 years-old, and residing in metropolitan areas. A total of 26 (44%) participants were living in Victoria, and 21 (35.6%) were residing in Queensland, making them the most represented states in the sample. In one case, a participant preferred not to disclose where they reside. Table 1 details the demographical characteristics of the sample.

#### Personal Interest in Climate Change

We also looked at personal interest in climate change, as we hypothesised that personal importance would be correlated with preparedness and engagement with mitigation strategies. A total of 56 (95%) participants reported being somewhat or very interested in climate change, while 45 (78%) participants said that climate change is very or extremely important to them personally. Personal importance was more prevalent in the age group 45–54 years-old (22%), followed by 35–44 years-old (22.3%). We found no correlation between personal interest and preparedness. 

### 3.2. Exposure to Climate-Health Impacts

#### 3.2.1. Workplace Affected by Climate Change

A total of 26 participants (44%) reported having their work affected by extreme weather events in the 12 months before the survey, as shown in Table 2. Of those participants who had their work affected in the past 12 months, a total of 14 (54%) are from the state of Queensland, and eight (30%) are from Victoria. Table 2 details the event that impacted participants’ workplaces in the twelve months prior to the survey.

#### 3.2.2. Exposure to Climate Change Impacts on Mental Health (Clients)

Participants were asked if their client’s health had been affected by climate change. This question was followed by an open question, in which participants could express all the health impacts. Increased stress and anxiety were the most common answers. Below are some of the participant responses: 


*“Increased stress, young people reporting distress about climate change and lack of political will. I had a client experience an exacerbation of their suicidal thoughts during bushfires as a result of distress at lack of climate change action.”*



*“Loss of property due to extreme weather events. Affects mental health, stress, family stability, and everything that goes with that (e.g., worse diets, burnout).”*



*“Asthma & respiratory-related impacts”*



*“Increased anxiety, uncertainty, stress. Other health impacts from social determinants—housing issues, cost of living.”*



*“Direct physical and mental health impacts from homes flooding; increased anxiety due to worry and stress; financial impacts of flooding increased anxiety and depression; temporary housing due to flooding. Understanding climate change-induced grief and sense of foreshortened future.”*


In the twelve months before the survey, 30 (50%) participants reported seeing clients with mental health impacted by climate change (Figure 1). A total of 17 (29%) respondents acknowledged they were unsure. Client mental health presentations are shown in Figure 1.

#### 3.2.3. Exposure to Climate Change Impacts on Mental Health (Personal)

Personal emotional impacts caused by climate change were reported by 52 (88%) participants (Figure 2). Those who reported climate change to be personally important were more likely to experience frustration (χ^2^ = 10.7, *df* = 1, *p* < 0.01), worry (χ^2^ = 8.4, *df* = 1, *p* < 0.01) and anxiety (χ^2^ = 11.9, *df* = 1, *p* < 0.01). 

### 3.3. Preparedness for Climate-Health Impacts

There was an overwhelming consensus among participants that more education and training opportunities (on the impacts of climate change on health) are necessary. A total of 50 (85%) participants would like to know more about the health impacts of climate change, while 49 (83%) agreed this should be covered in the curriculum for health-related professions; a total of 48 (81%) concurred it should be part of ongoing professional training, and 45 (76.8%) reported insufficient education and training opportunities. Table 3 shows participant knowledge about the impacts of climate change on health and their perception of available training. 

When asked about perceived preparedness, a total of 27 (46.8%) participants did not feel prepared to help clients presenting with health issues related to climate change; a total of 26 (44%) felt somewhat prepared, and six (10.1%) reported feeling very prepared. To test associations, we compiled perceived preparedness into two groups: unsure, not at all and not very prepared were classified as “not prepared”, and somewhat and very prepared were classified as “prepared”. As shown in Table 4, perceived preparedness was not associated with prior climate change training, knowledge of the health impacts of climate change, years of experience, and work being affected by climate change. However, we found a statistically significant association between perceived preparedness and awareness of professional statements, registration status, and the belief that general training to become a psychologist is sufficient. 

### 3.4. Willingness and Barriers to Acting on Climate Change

Despite 53 (90%) of respondents agreeing that climate change is a severe problem needing immediate action, a total of 50 (85%) believing the public need to be better informed about climate-health impacts, and 40 (68%) concurring that psychologists have a role in informing the public about it, we found that 39 (66%) participants do not communicate the health impacts of climate change to their clients. Table 5 shows the participant levels of comfort in climate change-related communications with their clients.

Ethics and lack of knowledge were noted as the main reasons for the lack of discussion of climate change with their clients, as 34 (58%) participants believe it is unethical to discuss climate change unless the topic is initiated by the client; we found 29 (49%) do not feel informed enough to communicate the health impacts of climate change, and noted a total of eight (4%) did not believe it was their responsibility. Some participants made use of the “other” option in the questionnaire to express their views: 


*“It is not the role of a practising psychologist to ‘educate’ their clients on climate change. Yes, we can help with the distress associated with it and provide clients with therapy and support to help manage that distress.”*



*“I don’t believe psychologists should be actively promoting their political ideals to clients. We need to remain neutral and non-biased so we can deal with clients’ problems competently and effectively with evidence-based research. I am not a climate scientist.”*


Participants were asked about discussions and climate action at their workplace. Although 45 (76%) of participants saw their peers as somewhat or very interested in the topic, only seven (11.8%) participants reported that climate change is often discussed at work, and 16 (27%) participants said their workplace is tackling climate change. Table 6 shows participant responses to climate action in the workplace. 

We found a statistical association between decision-making role and climate change being tackled at work. Those with no decision capacity and those who only decide on matters that impact them directly were more likely to report no climate actions at their workplace (χ^2^ = 4.33, *df* = 1, *p* = 0.03). Additionally, 14 (58%) participants who are fully responsible for decision-making in their organisation/team reported no tackling actions in their workplace. 

One participant shared how their workplace has developed policies for climate-driven events: 


*“We now have policies related to managing severe weather that includes clients and the community having a plan to keep services active. Fostering resilience is our priority”.*


Those who believe that climate change needs immediate action were more likely to report a lack of political will as a barrier in their workplace (χ^2^ = 14.85, *df* = 4, *p* < 0.01).

Despite only five (8.5%) participants stating they believe it is an ethical infringement to advocate for climate change, a total of 27 (46%) participants reported not feeling comfortable in doing so. We found that 33 (56%) participants are somewhat or extremely likely to advocate for climate action in their local community, observed 29 (49%) are somewhat or extremely likely to advocate for the development of a climate risk plan in their workplace, and noted 33 (56%) are somewhat or extremely likely to advocate for emission reduction in their workplace.

Overall, the participants reported similar results for knowledge, need for training and perceptions of climate change as fellow Australian healthcare professionals [23]. However, the two groups differ considerably in their opinion on how their workplace could help tackle climate change, as shown in Figure 3.

## 4. Discussion

Research on healthcare professionals’ experiences of, and attitudes towards, the complex interaction between climate change and health is still scarce. The participants in this study share an overwhelming consensus that climate change affects human health and the education they received is insufficient, which is consistent with previous research [15,23].

Australia is no stranger to disasters. In the twelve months preceding the survey, severe floods devastated regions and affected millions of people [31,32]. Almost half of the participants were affected by a climate-driven event in the year before our survey. Although 50% of the psychologists surveyed identified climate impacts on their client’s health, about 30% of our sample admitted they were unsure. When we consider the prevalence of climate-related mental health presentations in Australia [5,33], the strong agreement among participants that more education is needed, and the unwillingness of participants to engage clients in conversations about climate change, we are drawn to question if these results adequately represent the incidence of climate-related mental health. This question is even more pertinent when we pay attention to the fact that similar studies with healthcare professionals reported higher identification rates of mental health being impacted by climate change [15,23]. 

The majority of participants were not knowledgeable about the health impacts of climate change, which might explain the lower identification rate of clinical presentation of climate-related mental distress, especially when we consider the high number of people affected by the 2022 floods in Australia. Despite the reported lack of specific training and knowledge, many participants reported feeling prepared to identify and manage climate-related mental health distress. This is a problematic finding, as psychologists are expected to demonstrate adequate self-reflection that will guide their learning goals for continued professional development [34]. While we acknowledge that this is an emerging research field, with significant gaps and insufficiently applied research on prevention and interventions, the psychologist role in the climate crisis extends beyond clinical settings [3,24,25,26,35].

Discussions about including climate change in the curriculum of health professional education are not new. Still, evidence shows that its implementation is not in pace with the urgency the climate crisis requires [18]. Given that the demand for mental health services is expected to increase, training institutes and professional associations have an opportunity to step up their efforts in preparing psychologists to become active participants in managing the climate crisis. After all, psychologists have an ethical obligation to protect lives, and training institutes and professional associations have a moral responsibility to guide their members to become active role models in this crisis [24].

Notwithstanding intense recruitment efforts, the sample size is a major limitation of the present study. Possible explanations for the low response rate are survey fatigue, insufficient or flawed recruitment strategies, and Australian psychologists’ lack of interest in climate change. Our small sample size does not allow generalisation of the results. Nonetheless, our study helps clarify important aspects of the topic, and is an important step that will guide future research in the field. 

Our survey suggests that the Australian psychologists surveyed are less interested in, and engaged with, climate action than other Australian healthcare professionals [23]. Despite acknowledging the severity of the climate crisis, the participants believed it is unethical to discuss climate change unless the topic is initiated by the client, which may indicate that, in line with previous research [15], Australian psychologists see climate change as too politicised and sensitive to discuss in their professional setting. This highly politicised view of climate change might have influenced our sample size and it makes us question if our results would be the same had we used terms like “environmental sustainability”, “sustainable healthcare” and/or “planetary health”, which were previously used in similar research [36]. We believe this is an interesting hypothesis for future research. 

We argue, however, that healthcare professionals have an ethical duty to do no harm. The healthcare industry is responsible for between 1% and 5% of the world’s greenhouse emissions [16], and the Australian healthcare sector is by itself responsible for 7 % of the country’s emissions [37]. Given climate change’s negative effects on health, abstaining from efforts to prevent and mitigate climate change is not only counterproductive, but also goes against the latest paradigm of planetary health. Considering the extent of the threat posed by climate change, engaging in climate action and shedding light on its risks to health is the responsibility of health professionals [38]. By not engaging with the topic, psychologists could miss an excellent opportunity to identify the mental health impacts of climate change, adequately intervene, and foster resilience in their clients and communities.

Fortunately, our findings show that only a minority of participants (8%) believe that it is unethical to advocate for climate change and about half of the participants are somewhat or extremely likely to advocate for climate action to reduce the risks to the population’s health. This shows that framing climate change as a health issue facilitates the inclusion of the topic in the educational curriculum of healthcare professionals and increases the likelihood that Australian psychologists will embrace the cause and participate in the prevention and mitigation of climate change [18,39]. 

Our study adds to a growing literature on health professionals’ knowledge and experience of the relationship between climate change and mental health. Future research should focus on the efficacy of educational intervention programs aimed at healthcare professionals; the relationship between ethical concerns and psychologists’ engagement with the topic of climate change; and the roles assumed by professionals working exclusively in private practice when addressing public health issues. 

## 5. Conclusions

In the wider context of climate change, healthcare providers have been urged to step up to their responsibilities as health advocates and health promoters. Although advances have been made, there is significant room for improvement. Our survey showed that surveyed Australian psychologists are, despite acknowledging the importance of the climate crisis and its impacts on human health, not prepared for, and not engaged with, climate change. Lack of knowledge and ethical concerns, which were identified as the main barriers to engaging in communication, advocacy, and mitigation strategies, can be resolved through more education. Educational institutes and professional societies have an excellent opportunity to increase training, reframe climate change as a health crisis, and lead Australian psychologists to become active participants in climate action. 

## Figures and Tables

**Figure 1 ijerph-21-00218-f001:**
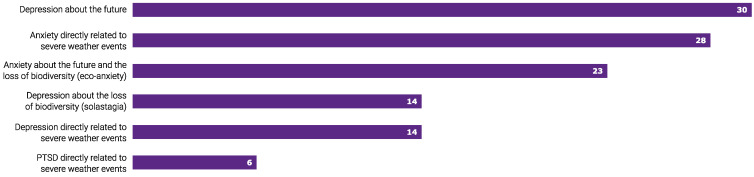
Climate change-related mental health presentation (*n* = 30) was observed by 59 surveyed Australian psychologists in the twelve months prior to the survey.

**Figure 2 ijerph-21-00218-f002:**
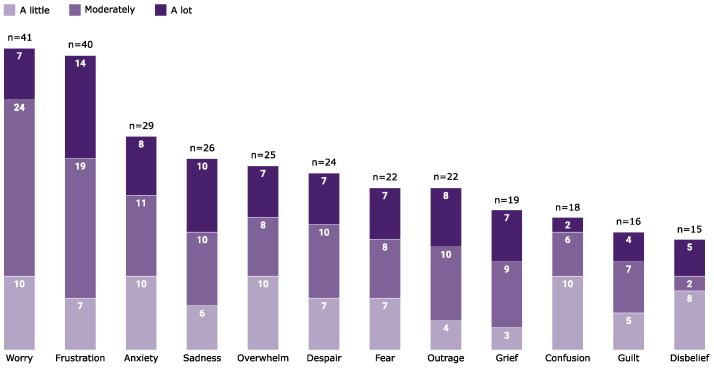
Surveyed Australian psychologists (*n* = 59) own emotional experience of climate change.

**Figure 3 ijerph-21-00218-f003:**
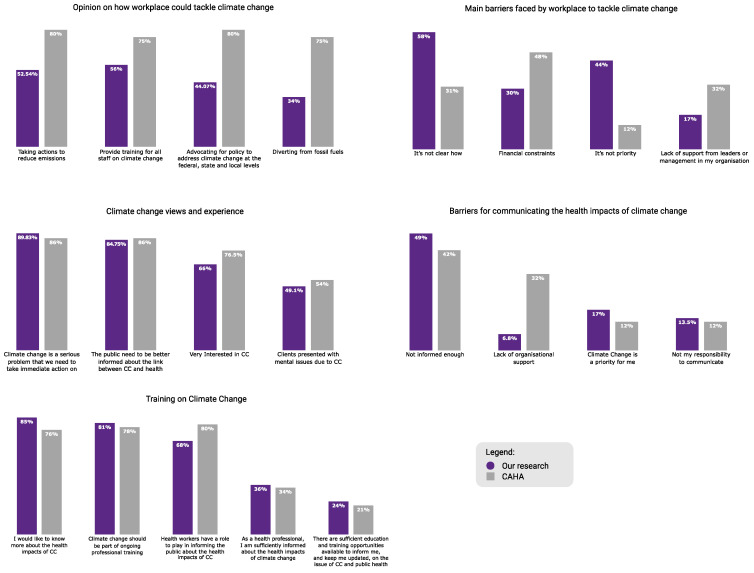
Overall comparison of the findings from the Climate and Health Alliance survey (*n* = 875) of physicians, nurses, midwives, public health professionals and medical students, and the present research of Australian psychologists’ (*n* = 59) perceptions and experiences of climate change.

**Table 1 ijerph-21-00218-t001:** Demographical characteristics of participants (*N* = 59).

Demographical Variables		*N* (%)
	18 to 24 years-old	2 (3.4%)
	25 to 34 years	12 (20.4%)
**Age**	35 to 44 years-old	21 (35.6%)
	45 to 54 years-old	15 (25.5%)
	55 to 64 years-old	4 (6.8%)
	Man	8 (13.6%)
**Gender**	Woman	49 (83%)
	Transgender	1 (1.7%)
	Prefer not to say	1 (1.7%)
	ACT	1 (1.7%)
	NSW	5 (8.5%)
	QLD	21 (35.6%)
**State**	SA	1 (1.7%)
	TAS	1 (1.7%)
	VIC	26 (44.1%)
	WA	3 (5%)
	Missing data	1 (1.7%)
	Less than 5	19 (32.2%)
**Experience**	5 to 10 years	14 (23.7%)
	10 to 15 years	10 (17%)
	More than 15 years	16 (27%)
	Provisional	13 (22%)
**Registration Type**	General registration (without endorsement)	19 (32.2%)
	General (with endorsement)	27 (45.8%)
	Clinical psychology	18 (30.5%)
	Neuropsychology	2 (3.4%)
**Endorsement Type**	Health psychology	3 (5.0%)
	Counselling psychology	2 (3.4%)
	Organisational psychology	2 (3.4%)
	No endorsement	32 (54.2%)
	Private practice	36 (61%)
	Government	4 (6.8%)
	Community mental health Services	5 (8.5%)
	University	5 (8.5%)
**Professional**	Hospitals	3 (5.1%)
**Setting**	School	3 (5.1%)
	Aboriginal health services	1 (1.7%)
	Other private organisations	1 (1.7%)
	Other	1 (1.7%)
	Direct client contact	56 (95%)
**Roles**	Research	13 (22%)
	Academic	10 (17%)
	Leadership	7 (12%)

**Table 2 ijerph-21-00218-t002:** Participant’s workplace exposure to climate- change driven events (*N* = 26).

Extreme Weather Event	*N* (%)
Bushfires smoke	1 (1.7%)
Heat	4 (6.8%)
Flooding	19 (32.2%)
Storms	13 (22%)

Note: more than one answer was allowed.

**Table 3 ijerph-21-00218-t003:** Participant knowledge and perception of available training on climate–health impacts (*N* = 59).

		*N* (%)
	Definitely affects human health	41 (69.4%)
	Probably affects human health	14 (23.7%)
**Climate Change**	Might affect human health	3 (5%)
	Probably does not affect human health.	1 (1.7%)
**Health Impacts**	I feel well or very informed	19 (32.2%)
**of Climate Change**	I feel somewhat or not informed	40 (68%)
	Would like to know more about the health impacts of climate change	50 (84.7%)
**Knowledge of Climate Change**	Should be covered in the curriculum for health-related professions	49 (83%)
	Should be part of ongoing professional training	48 (81%)
	It is not insufficiently covered in the current education and training opportunities.	45 (76.8%)
	Never had any formal training on climate change and its impact on health	43 (73%)
	There is not enough training	45 (76.3%)
**Training on Climate Change**	Not familiar with any professional association’s position statement on climate change	37 (67.7%)
	Somewhat or strongly agreed that the training received as a psychologist gives sufficient information about the health impacts of climate change.	21 (36%)

**Table 4 ijerph-21-00218-t004:** Statistical association of participant’s perceived preparedness for climate–health impacts (*N* = 59).

	Prepared *N* (%)	Not Prepared *N* (%)	Test Statistics
**Climate Change Training**			
Training	11 (34.4%)	5 (18.5%)	χ^2^ (1) = 1.862 *p* = 0.1
No formal training	21 (65.6%)	22 (81.5%)	
**Health Impacts of Climate Change**			
Well or very informed	12 (37.5%)	7 (25.9%)	χ^2^ (1) = 0.895 *p* = 0.3
Somewhat or not informed	20 (62.5%)	20 (74.1%)	
**Professional Experience**			
Up to 9 years	16 (50%)	17 (63%)	χ^2^ (1) = 0.99 *p* = 0.31
10 + years	16 (50%)	10 (37%)	χ^2^ (1) = 0.99 *p* = 0.31
**Awareness of Professional Statements**			
Somewhat familiar	16 (50%)	6 (22.2%)	χ^2^ (1) = 4.83 *p* = 0.02
Not familiar	16 (50%)	21 (77.8%)	χ^2^ (1) = 4.83 *p* = 0.02
**Registration Status**			
Registered psychologist	28 (87.5%)	18 (66.7%)	χ^2^ (1) = 3.69 *p* = 0.05
Provisional psychologist	4 (23.5%)	9 (33.3%)	χ^2^ (1) = 3.69 *p* = 0.05
**Sufficiently Informed as a Psychologist**			
Psychology training is sufficient	17 (53.1%)	4 (14.8%)	χ^2^ (1) = 9.37 *p* = 0.02
Psychology training is not sufficient	15 (56.9%)	23 (85.2%)	χ^2^ (1) = 9.37 *p* = 0.02
**Work Affected by Climate Change**			χ^2^ (1) = 0.99 *p* = 0.31
Yes	16 (50%)	10 (41.2%)	χ^2^ (1) = 0.99 *p* = 0.31
No	16 (50%)	17 (58.8%)	

**Table 5 ijerph-21-00218-t005:** Participants’ levels of comfort when communicating with their clients about climate change (*N* = 59).

		*N* (%)
**Communicating the health impacts of climate change to clients**	felt somewhat or very comfortable	29 (49%)
	not comfortable	19 (32%)
**Communicating how their clients can protect themselves from the health effects of climate change**	somewhat or very comfortable	27 (45%)
	not comfortable	25 (42%)
**Communicating about actions and day-to-day changes people in the broader community can make to prevent climate change from worsening**	were comfortable	25 (43%)
	not comfortable	25 (43%)

**Table 6 ijerph-21-00218-t006:** Participant views on climate action in the workplace (*N* = 59).

		*N* (%)
**As a way to tackle climate change, do you think your workplace could be:**	Taking actions to reduce emissions	31 (52.5%)
	Advocating for policy to address climate change at the federal, state and local levels	26 (44%)
	Diverting from fossil fuels	20 (33.9%)
	Provide training for all staff on climate change	33 (55.9%)
	Providing information/ help the public to change behaviour towards acting on climate change	33 (55.9%)
	Providing support to the population to cope with climate change	36 (61%)
	I do not believe it is my organisation’s role to tackle climate change	10 (17%)
**What barriers does your workplace face in tackling climate change?**	It is not clear how my organisation can tackle climate change	34 (57.6%)
	It is not a priority	26 (44%)
	Financial constraints	18 (30.5%)
	Lack of knowledge	34 (58%)
	Lack of time	17 (29%)
	political will	13 (22%)

## Data Availability

Data collected in this survey are available upon reasonable request to Gabriela Stilita (g.santosstilitacardoso@uq.net.au).

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
