# Peer review of "Keeping Sane in a Changing Climate: Assessing Psychologists’ Preparedness, Exposure to Climate-Health Impacts, Willingness to Act on Climate Change, and Barriers to Effective Action"

_ijerph, 2024, doi:10.3390/ijerph21020218_

Round 1

Reviewer 1 Report

Comments and Suggestions for Authors

The article "Keeping sane on a changing climate: assessing Psychologists’ 2 preparedness, exposure to climate-health impacts, and willing-3 ness and barriers to acting on climate change" by Stilita and Charlson represents a pressing issue related to the impact of climate change, that is on mental health. The article is well written. 

I have the following minor comments on the article which need to be addressed by the authors:

1. The sample size looks low. There is no mention of the sample size requirement in the article. Also, there is no mention of an approximate number of registered psychologists working in the study area. 

2. Line 190: "......impacted by climate change; see Fig 1": Please change to "....impacted by climate change (Fig 1)."

3. Line 195-196: ".......by 52 (88%) participants, see Fig. 2": Please change to ".....by 52 (88%) participants (Fig. 2)."

4. Please check the texts used for all the figures. The text needs to be more clear for better readability.

Author Response

Response to reviewer 1: Thank you very much for taking the time to review our article. We agree with your comments and the following modifications were made.

Comments:

  1. The sample size looks low. There is no mention of the sample size requirement in the article. Also, there is no mention of an approximate number of registered psychologists working in the study area.

Response to comment 1: The sample size was indeed a limitation, to the point that the survey was extended from what was originally planned. We have accepted your comment and added the number of registered psychologists at the time of the survey “Between July and September 2022, there were 35,315 general registered psychologists and 7.977 provisional psychologists, totalising 43.292 professionals of interest [28].” [lines 125-127]. And “The number of psychologists who had access to the recruitment ads is unknown. Every respondent that started the survey, has  finished it, resulting in a 100% completion rate.” [lines 131-133]

  1. Line 190: "......impacted by climate change; see Fig 1": Please change to "....impacted by climate change (Fig 1)."
  2. Line 195-196: ".......by 52 (88%) participants, see Fig. 2": Please change to ".....by 52 (88%) participants (Fig. 2)."

Response to comments 2 and 3: Thanks for pointing this. We have changed the manuscript as requested, which improved readability, and now reads: “impacted by climate change (Fig 1).” [lines 216], and "by 52 (88%) participants (Fig. 2)." [lines 233]

  1. Please check the texts used for all the figures. The text needs to be more clear for better readability.

Response to comment 4: Figure 1. Climate change-related mental health presentations observed by 59 surveyed Australian psychologists, in the twelve months prior to the survey. [lines 221 – 222]. Figure 2. Surveyed Australian psychologists (n=59) own emotional experience with climate change. [line 229]. Figure 3. Overall comparison of the findings from the Climate and Health Alliance survey (n = 875 among physicians, nurses, midwives, public health professionals and medical students) and the present research on Australian psychologists (n=59) perceptions and experiences on climate change. [lines 303 – 306]

Reviewer 2 Report

Comments and Suggestions for Authors

Dear authors,

Thank you for the opportunity to review your manuscript. I applaud your efforts to advance this work among colleagues in psychology and consider this extremely important work.

I have only two small suggestions for amendments:

First, in the discussion, I suggest moving your comments about sample size (lines 334-338) before line 310. The statement in 310-311 is overly generalized considering what you rightly state about your sample size. If you place the former before the latter, you could then modify the statement in lines 310-311 to be more narrowly representative of your study.

Second, it seems to me that it would be helpful to add a paragraph in your introduction that looks beyond the narrow use of the term ‘climate change’. Climate change is an undoubtedly central issue, but one in an exponentially broadening and deepening field touching on concepts like planetary health, One Health, Ecosystems Health, sustainability, sustainable development, environmental sustainability, biodiversity loss, etc etc. I have been involved in a survey study (https://www.mdpi.com/1660-4601/19/16/10121) using ‘environmental sustainability’ as its focus term, but encompassing climate change, and more, and I believe there have been several others with a slightly different terminological focus but highly resonant subject matter (we also point to some in our article). I think pointing to the broader body of evidence building here will be helpful in building your argument. The point then, would be to provide a brief overview the broader discourse that ‘climate change’ is embedded in, and the broader body of evidence to which your study is contributing. You could then also pick this up briefly in your discussion.

Kind regards,

Author Response

Response to Reviewer 2

Response to reviewer 2: Thank you very much for taking the time to review our article. We agree with your comments and the following modifications were made:

Comments:

  1. First, in the discussion, I suggest moving your comments about sample size (lines 334-338) before line 310. The statement in 310-311 is overly generalized considering what you rightly state about your sample size. If you place the former before the latter, you could then modify the statement in lines 310-311 to be more narrowly representative of your study.

Response to comment 1: This is a great suggestion, thank you. We have adopted the change suggested. We moved the paragraph about sample size as suggested [ lines 337 – 341], we reinforced our recruitment efforts (“Notwithstanding intense efforts on recruitment, the sample size is a major weakness of the present study.”) [lines 337-338], and we added the word “surveyed” (“Our survey shows that the Australian psychologists surveyed are”) in line 343 to improve readability and to reinforce that our results are not generalizable. 

  1. Second, it seems to me that it would be helpful to add a paragraph in your introduction that looks beyond the narrow use of the term ‘climate change’. Climate change is an undoubtedly central issue, but one in an exponentially broadening and deepening field touching on concepts like planetary health, One Health, Ecosystems Health, sustainability, sustainable development, environmental sustainability, biodiversity loss, etc etc. I have been involved in a survey study (https://www.mdpi.com/1660-4601/19/16/10121) using ‘environmental sustainability’ as its focus term, but encompassing climate change, and more, and I believe there have been several others with a slightly different terminological focus but highly resonant subject matter (we also point to some in our article). I think pointing to the broader body of evidence building here will be helpful in building your argument. The point then, would be to provide a brief overview the broader discourse that ‘climate change’ is embedded in, and the broader body of evidence to which your study is contributing. You could then also pick this up briefly in your discussion.

Response to comment 2: Thank you very much for sharing the survey you have been involved with. As a result of your thoughtful comment, we have improved our introduction to explore a bit more the complex interactions between climate change and health through a planetary health approach, which now reads: When we consider the complex interaction between socioeconomical determinants of health and exposure to climate change, it becomes clear that the full impact of climate change on people’s mental health has not been fully understood yet. It has been argued that only by adopting a system thinking approach [10], we will be able to identify the intricated relationship between the various vulnerability factors and different exposures pathways that result in poorer mental health caused by climate change [2, 11, 13]. As climate change worsens, it is sensible to expect higher mental health presentation as a result of the increased incidence of the physical health conditions caused by climate change, awareness of the environmental degradation, forced displacement, community disruption, nutritional depletion, increased poverty, among others. Thus, with the escalation of climate change, the demand for mental health services is expected to worsen [3].

Notwithstanding the severe impacts on health caused by climate change, the perceived role of health professionals in climate change has not been the subject of many studies. Evidence shows that despite health professionals agreeing that climate change impacts health, they have insufficient information on the link between climate change and health [14,15].

The healthcare sector is a huge contributor to climate change. Directly or indirectly, the sector is responsible for between 1% and 5% of the world’s greenhouse emissions [16]. The healthcare sector footprint has triggered recent calls for a paradigm change on how we understand health, the inclusion of environmental health into healthcare education. and the healthcare professionals’ responsibility to address how they related with climate change on their professional practice [17 - 19]. The planetary health paradigm understands the direct link between human health and environmental health, framing climate change as a planetary as well as a human health crisis [19 - 21]. This shift towards the planetary health paradigm and the urgent need to address climate change and its forecasted pressure on healthcare systems has led the World Health Organisation to recently publish its framework for a climate resistant and low carbon health system, highlighting the need for a climate-smart health workforce [22].” [lines 38 – 64].

As suggested, we return to the concept of planetary health in our discussion: Given climate change negative effects on health, abstaining from the efforts to prevent and mitigate climate change is counterproductive, but also go against the latest paradigm of planetary health.” [lines 352 – 354]. We also added the following paragraph: “This politicised perception of climate change makes us question if our results would be the same had we used terms like environmental sustainability, sustainable healthcare or planetary health, as used in similar research [38].” [360 – 362]

Reviewer 3 Report

Comments and Suggestions for Authors

Dear colleagues,

although the topic is for sure of interest, I think that a sample as small as N=59 is not sufficient to publish on a scientific journal such as IJERPH. This is the case also in light of the population from which the sample was drawn, which is of more than 27000 psychologists (please see here: https://psychology.org.au/about-us/news-and-media/media-releases/2022/unpaid,-underfunded-and-overworked-psychologists-o). I undertand that authors provide some justification and limitations for the small sample size, but in the end this does not affect the substance of the study. Specifically, sample size is so small that trying to derive any kind of provisional conclusions (not to say generalizations) is very difficult. I don't know how such issue can be dealt with in a potential revision; my suggestion is to try to put together a sample of at least 250/300 participants, which is not VERY big, still it would be more in line with the idea of gaining an initiatl insight into an under-researched topic.

I'm also curious to know, for comparison, what are the number of participants of the survey that investigated the same issues related to climate change in other health professions, referred by authors on p. 2, line 52. 

Introduction. I found some ambiguity on the main aim of the study, since it seems that the study investigates not only knowledge of psychologists about the mental health implications of climate change, but also the willingness of psychologists to take active action on climate change issues. How can the two issues go together? Why they have been put together? I don ot feel that this was explained very well, particularly as regards the willingness of psychologists to take active action on climate change issues as part of their professional role. Why psychologists should do this as part of their professional role? I'm not saying this is not potentially important, but that this is not explained very well in the Introduction.  

p. 2, line 67. Introduction. It is not very clear what is the aim of the study described in the following way:  "Assess factors associated with the findings of the study aims detailed above".

Please have a look at this, which may be of help in better articulating the role of psychology on climate change: https://www.annualreviews.org/doi/10.1146/annurev-psych-032720-042905

Good luck with developing the paper.

Comments on the Quality of English Language

Generally, English is ok. But I'm not an English native speaker.

Author Response

Response: Thank you very much for taking the time to review our article. 

Comment 1: Although the topic is for sure of interest, I think that a sample as small as N=59 is not sufficient to publish in a scientific journal such as IJERPH. This is the case also in light of the population from which the sample was drawn, which is of more than 27000 psychologists (please see here: https://psychology.org.au/about-us/news-and-media/media-releases/2022/unpaid,-underfunded-and-overworked-psychologists-o). I understand that authors provide some justification and limitations for the small sample size, but in the end this does not affect the substance of the study. Specifically, sample size is so small that trying to derive any kind of provisional conclusions (not to say generalizations) is very difficult. I don’t know how such an issue can be dealt with in a potential revision; my suggestion is to try to put together a sample of at least 250/300 participants, which is not VERY big, still, it would be more in line with the idea of gaining an initial insight into an under-researched topic.

I’m also curious to know, for comparison, what are the number of participants of the survey that investigated the same issues related to climate change in other health professions, referred by authors on p. 2, line 52.

Response to comment 1: We agree that the sample size was indeed an important problem to the point that the survey was extended from what was originally planned. We have added to the manuscript the official number of registered psychologists at the time of the survey, which reinforce our claims that our results are not generalizable. “Between July and September 2022, there were 35,315 general registered psychologists and 7.977 provisional psychologists, totalising 43.292 professionals of interest [28].” [lines 125-127]. And “The number of psychologists who had access to the recruitment ads is unknown. Every respondent that started the survey has finished it, resulting in a 100% completion rate.” [lines 131-133].

We politely disagree with your comment that a study with such a sample size should not be published on IJERPH, as other studies on climate change with lower sample sizes have been previously published by IJERPH[1],[2]. To the best of our knowledge, this is the first study worldwide assessing psychologists’ personal and professional experiences on climate change, and it lays an important foundation for future research despite the impossibility of generalisation of the findings. Furthermore, in their discussion on sample size, Memon et al. (2020) [3] state that sample sizes are influenced by a variety of factors, including research approach, analytical method, and sample size for similar studies (among others). Although similar studies have a higher sample size, their population of interest are multi-professional, and none of them include psychologists. We also argue that our study is exploratory, and where associations were calculated, we used simple statistical analysis, which, according to Memon et al., a sample between 50 and 100 samples is generally enough. In her study on power analysis and exploratory research, Haile (2023)[4] argues that given the lack of a pre-defined hypothesis, exploratory studies don’t require power analysis and low sample sizes are common. Considering our intense effort in participant recruitment and some of the answers we received, we believe it is possible that our low sample size is a reflection of the highly politicized view of climate change, which by itself is an interesting hypothesis for future research. We edited the manuscript to reflect this hypothesis “This highly politicized view of climate change might have influenced our sample size, which by itself is an interesting hypothesis for future research.“ [lines 352- 354]  

Comment 2: Introduction. I found some ambiguity on the main aim of the study, since it seems that the study investigates not only knowledge of psychologists about the mental health implications of climate change, but also the willingness of psychologists to take active action on climate change issues. How can the two issues go together? Why they have been put together? I don ot feel that this was explained very well, particularly as regards the willingness of psychologists to take active action on climate change issues as part of their professional role. Why psychologists should do this as part of their professional role? I’m not saying this is not potentially important, but that this is not explained very well in the Introduction. 

  1. 2, line 67.

Response to comment 2: We appreciate your comment on your perceived ambiguity on the aims of the present study. As a psychologist myself (GS), I often get questioned if psychologists should be involved in climate change in their professional practice. As a result of your comments, we have improved our introduction to better explore the concepts of planetary health and the recent calls for all healthcare professionals to be active participants in climate actions: When we consider the complex interaction between socioeconomic determinants of health and exposure to climate change, it becomes clear that the full impact of climate change on people’s mental health has not been fully understood yet. It has been argued that only by adopting a system thinking approach [10], we will be able to identify the intricated relationship between the various vulnerability factors and different exposure pathways that result in poorer mental health caused by climate change [2, 11, 13]. As climate change worsens, it is sensible to expect higher mental health presentation as a result of the increased incidence of physical health conditions caused by climate change, awareness of environmental degradation, forced displacement, community disruption, nutritional depletion, and increased poverty, among others. Thus, with the escalation of climate change, the demand for mental health services is expected to worsen [3].

Notwithstanding the severe impacts on health caused by climate change, the perceived role of health professionals in climate change has not been the subject of many studies. Evidence shows that despite health professionals agreeing that climate change impacts health, they have insufficient information on the link between climate change and health [14,15].

The healthcare sector is a huge contributor to climate change. Directly or indirectly, the sector is responsible for between 1% and 5% of the world’s greenhouse emissions [16]. The healthcare sector footprint has triggered recent calls for a paradigm change in how we understand health, the inclusion of environmental health into healthcare education, and the healthcare professional’s responsibility to address how they relate to climate change in their professional practice [17 - 19]. The planetary health paradigm understands the direct link between human health and environmental health, framing climate change as a planetary as well as a human health crisis [19 - 21]. This shift towards the planetary health paradigm and the urgent need to address climate change and its forecasted pressure on healthcare systems has led the World Health Organisation to recently publish its framework for a climate-resistant and low-carbon health system, highlighting the need for a climate-smart health workforce [22].

[lines 38 – 64].

Comment 3: Introduction. It is not very clear what is the aim of the study described in the following way: “Assess factors associated with the findings of the study aims detailed above”.

Response to comment 3: Thank you for your comment on our aim number 4. We agree that it lacks clarity. Our intention with that aim was to make it clear that we would run some exploratory statistical analysis that could help us understand factors associated with psychologists’ preparedness, exposure and wiliness to act on climate change. We decided to delete aim 4 and edit the manuscript to make our point clear: “Thus, the present study aims to identify and analyse factors associated with Australian psychologists” [line 83]. 

Comment 4: Please have a look at this, which may be of help in better articulating the role of psychology on climate change: https://www.annualreviews.org/doi/10.1146/annurev-psych-032720-042905

Response to comment 4: Thank you for referring us to Steg's (2023) article. We are familiar with this important work, and we value its reflection on the varied values that contribute to a person’s engagement in climate action. While this is indeed an important study, as healthcare professionals, psychologists are guided by ethical guidelines and institutional/ governmental policies. Several psychology boards published statements on climate change and the psychologists’ role in climate action. As an example, the British Psychological Society urges its members to be role models and cites the example of members who participate in groups like Extinction Rebellion, XR Psychologists, Medact, PsychDeclares, Climate Psychology Alliance, among others (https://www.bps.org.uk/member-networks/division-clinical-psychology/climate-change). Additionally, as stated above, WHO has published different frameworks aiming to develop a climate-active healthcare workforce.

Response to Comments on the Quality of English Language: The final manuscript has been through extensive English revision.

[1] Chau, J.Y.; Dharmayani, P.N.A.; Little, H. Navigating Neighbourhood Opposition and Climate Change: Feasibility and Acceptability of a Play Street Pilot in Sydney, Australia. Int. J. Environ. Res. Public Health202320, 2476. https://doi.org/10.3390/ijerph20032476

[2] Gunasiri, H.; Wang, Y.; Watkins, E.-M.; Capetola, T.; Henderson-Wilson, C.; Patrick, R. Hope, Coping and Eco-Anxiety: Young People’s Mental Health in a Climate-Impacted Australia. Int. J. Environ. Res. Public Health 202219, 5528. https://doi.org/10.3390/ijerph19095528

[3] Memon, Mumtaz & Ting, Hiram & Hwa, Cheah & Ramayah, T. & Chuah, Francis & Cham, Tat-Huei. (2020). Sample Size for Survey Research: Review and Recommendations. Journal of Applied Structural Equation Modeling. 4(2). i-xx. 10.47263/JASEM.4(2)01.

[4] Haile ZT. Power Analysis and Exploratory Research. Journal of Human Lactation. 2023;39(4):579-583. doi:10.1177/08903344231195625

Round 2

Reviewer 3 Report

Comments and Suggestions for Authors I read the revised version of the MS. and the responses given to my comments. Although it is clear that authors have tried their best, I'm against publication because the sample size is absolutely too small to gain any kind of useful information from the study. Deontologically speaking, I can't give a different assessment. I've also looked at the references quoted by authors (e.g., Menon et al), but I'm not convinced that a sample of 59 from a population of more than 30.000 is enough. Perhaps the present article could be framed in terms of a pilot investigation, but in that case I'm not sure it should be published on a scientific journal such as IJERPH. Recently a longitudinal study of mine has been DESK rejected because it was based on 89 participants x 3 waves of data collection. In other words it was a longitudinal study. The present study has a lower sample size and it is cross-sectional. Additionally, it is not a qualitative study for which low sample sizes may be accepted.
I'm very sorry for this negative evaluation. All the best for further developing your work.

Author Response

Once again, we thank the reviewer for taking the time to review our revised manuscript.

Comment 1: Although it is clear that the authors have tried their best, I’m against publication because the sample size is absolutely too small to gain any kind of useful information from the study.

Response to Comment 1:

We disagree with the reviewer’s opinion that our manuscript should not be published based on its sample size. It is clear throughout the manuscript that there is no intention to generalise our findings, as this is a novel exploratory study. As such, our results have pointed to many directions for future research, which, in our opinion, is very useful.

Comment 2: Deontologically speaking, I can’t give a different assessment. I’ve also looked at the references quoted by authors (e.g., Menon et al), but I’m not convinced that a sample of 59 from a population of more than 30.000 is enough.

Response to Comment 2: Although the total population is important for determining sample size, it is not the only aspect of sample size calculation. Effect size, power, significance level, and variability effect should be considered when calculating sample size that would allow for generalising the findings. The manuscript clearly shows that we do not intend to make a statistical inference from our findings. The study is fully exploratory, and there is no hypothesis to test. Furthermore, it is disappointing that despite the provision of evidence showing the adequacy of our sample size for this type of study and prior IJERPH publication of studies on climate change with smaller sample sizes, the reviewer adopted a moral obligation that is not evidence-based to refuse publication.

Comment 3: All the best for further developing your work.

Response to Comment 3: While the reviewer expressed their best wishes for developing the present manuscript, no suggestion for possible improvement was made. Moreover, the reviewer has downgraded its initial manuscript evaluation despite acknowledging that we “tried our best”. While in the first round of reviews, the manuscript’s references, description of methods, presentation of results and conclusion were assessed as adequate, only the introduction and the references received the same evaluation in the second round. Not only do we disagree that the modifications made during revision negatively impacted the manuscript, but we also add that no change was made to the results section (apart from the improvement of the Figures descriptors, requested by another reviewer). There is no justification for the reviewer’s updated evaluation. We stand by our position that the research designer is adequate for this type of study.